# Pathogenicity Factors of Botryosphaeriaceae Associated with Grapevine Trunk Diseases: New Developments on Their Action on Grapevine Defense Responses

**DOI:** 10.3390/pathogens11080951

**Published:** 2022-08-22

**Authors:** Marie Belair, Alexia Laura Grau, Julie Chong, Xubin Tian, Jiaxin Luo, Xin Guan, Flora Pensec

**Affiliations:** 1Laboratoire Universitaire de Biodiversité et Écologie Microbienne, University Brest, INRAE, F-29280 Plouzané, France; 2Unité de Recherche Vigne et Oenologie Colmar-Reims, UR3991, Université de Haute Alsace, 33 Rue de Herrlisheim, F-68000 Colmar, France; 3College of Horticulture and Landscape Architecture, Southwest University, Chongqing 400716, China; 4Key Laboratory of Horticulture Science for Southern Mountainous Regions, Ministry of Education, Chongqing 400716, China

**Keywords:** Botryosphaeriaceae, dieback, grapevine, trunk diseases, pathogenicity factors, plant defense response

## Abstract

Botryosphaeriaceae are a family of fungi associated with the decay of a large number of woody plants with economic importance and causing particularly great losses in viticulture due to grapevine trunk diseases. In recent years, major advances in the knowledge of the pathogenicity factors of these pathogens have been made possible by the development of next-generation sequencing. This review highlights the knowledge gained on genes encoding small secreted proteins such as effectors, carbohydrate-associated enzymes, transporters and genes associated with secondary metabolism, their representativeness within the Botryosphaeriaceae family and their expression during grapevine infection. These pathogenicity factors are particularly expressed during host–pathogen interactions, facilitating fungal development and nutrition, wood colonization, as well as manipulating defense pathways and inducing impacts at the cellular level and phytotoxicity. This work highlights the need for further research to continue the effort to elucidate the pathogenicity mechanisms of this family of fungi infecting grapevine in order to improve the development of control methods and varietal resistance and to reduce the development and the effects of the disease on grapevine harvest quality and yield.

## 1. Introduction

The Botryosphaeriaceae family (Botryosphaeriales, Ascomycetes) includes species with a worldwide distribution and a wide host range [1,2,3,4]. To date, around 22 genera and 300 species have been identified in this family [5,6], characterized as saprotrophic, endophytic or pathogenic and sometimes these trophic regimes are characteristic of the same species [7,8,9]. Pathogenic species can enter the host plant through wounds or natural openings [10], they settle latently in the host tissues and become pathogenic when the plant is stressed [9,11,12].

The symptoms are often described on perennial plants such as apples, blueberries, citrus, grapes, conifers or eucalyptuses; and range from dieback, to cankers, fruit rots and leaf discolorations [13,14,15,16]. Among the affected plants, Botryosphaeriaceae are well known as involved in the grapevine trunk diseases (GTD). Grapevine, whose physiology and composition of the berries is the subject of various studies [17,18,19], has been the most economically impacted plant of agronomic interest in recent decades [20,21] due to yield reductions, increased production costs and shorter vineyard life span [22,23].

In recent years, the development of next-generation sequencing techniques has led to major advances in the understanding of the virulence mechanisms of these pathogens [5,24,25,26,27,28]. Thus, different classes of genes involved in fungal pathogenesis have been identified as belonging to the arsenal of Botryosphaeriaceae, such as those encoding effectors and other small proteins, enzymes associated with wood degradation, peptidases, transporters and those associated with secondary metabolism [5,10,26,27]. Botryosphaeriaceae toxins, secondary metabolites and wood degradation enzymes are known to primarily exert a destructive effect on grapevine cells and are hypothesized to be the cause of the observed symptoms (wood and bud necrosis, foliar discolorations) [29]. However, recent work also suggests that Botryosphaeriaceae effectors could have a more subtle role than cell and tissue damaging, by allowing fungi to escape from grapevine immunity [30].

The objective of this review is to highlight the latest knowledge established in the literature on the pathogenicity factors of the main agents associated with Botryosphaeria dieback in grapevines such as *Diplodia seriata*, *Diplodia mutila*, *Lasiodiplodia theobromae*, *Botryosphaeria dothidea*, *Neofusicoccum parvum*, *Neofusicoccum mediterraneum* or *Dothiorella viticola*. The main classes of factors will be deciphered and their dispersion as well as their specificities within the main species will be highlighted. From the view of gene regulation and morphology modification, the effects of these pathogenicity factors on defense reactions and metabolism of grapevine will then be discussed and will allow to better understand the interaction between grapevine and Botryosphaeriaceae with regard to the research advances of these last years.

## 2. Pathogenicity Factors of Botryosphaeriaceae Species Associated with Grapevine Trunk Diseases

### 2.1. Small Secreted Proteins as an Important Element of Plant–Fungi Interaction

Botryosphaeriaceae species are known to cause foliar symptoms but, to date, they have never been isolated from leaves or fruits. Thus, it has been postulated that these symptoms are due to phytotoxins and secreted proteins (SP) produced by the fungi in the perennial part of the plant and then translocated to the leaves through the transpiration stream [31,32]. Moreover, for many pathogens, particularly in fungi and oomycetes, the excretion of proteins is essential for pathogenicity [33].

One category of SP seems of particular interest: the small secreted proteins (SSP). Often defined as proteins of less than 300 amino acids, they contain a signal peptide that permits secretion. These SSPs play an important role in the plant–fungal interaction and in pathogen virulence mechanisms [34]. Few studies have been conducted on these effectors in the Botryosphaeriaceae family, but Yu et al. (2022) reported that the number of SP widely varies between genera of Botryosphaeriaceae [33]. Species belonging to the *Macrophomina*, *Botryosphaeria* and *Neofusicoccum* genera are described as containing between 922 and 1118 SPs compared to only 671 to 866 SP in the genera *Diplodia*. In the same study, the authors indicate that two Botryosphaeriaceae genera (*Macrophomina* and *Botryosphaeria*) may be able to release over 250 SSPs while all *Diplodia* species considered produce less than 200 SSPs [33]. The most studied small effectors are probably the small cysteine-rich secreted proteins (SCSP). Several of these SCSPs have been identified as virulence factors in fungi and oomycetes. SCSPs can inhibit extracellular plant proteases, protect the hyphae against chitin-triggered defenses or play a role in programmed cell death (PCD) mechanisms [34,35]. These functions have never been highlighted in Botryosphaeriaceae species, nevertheless, Vázquez-Avendaño and colleagues (2021) have identified five new cysteine-rich proteins in *N. parvum* [35]. These proteins can build disulfide bonds and present a signal peptide. None of these five proteins have an identified function but, interestingly, two of them contain a CFEM domain [35]. This pattern unique to fungi and characterized with the presence of eight cysteines is predicted to be more abundant in pathogenic fungi, and has thus been identified as a putative virulence factor [36].

The LysM domain is another pattern largely described to be present in SSPs involved in pathogenic interactions between plants and fungi by protecting the hyphae from host defenses and preventing the recognition of the pathogen by the plant [37,38,39]. Although these mechanisms have been largely described in many plant–fungi interactions, less is known about their role in Botryosphaeriaceae pathogenicity. Harishchandra and colleagues, 2020, have identified three putative LysM-containing SSPs. Suppression by RNAi of one of them, LtLysM, resulted in fewer lesions on a susceptible grapevine cultivar, demonstrating the involvement of LtLysM1 in *L. theobromae* pathogenicity [37]. Other SSPs are suspected to contribute to the virulence of *Botryosphaeriaceae* species by suppressing plant immune defenses. In particular, Zhang et al. (2021) highlighted candidate effectors with a signal peptide capable of totally suppressing the BAX (Bcl-2–associated X) mediated PCD [30].

Among SSPs, the Nep1-like protein (NLP) family is widely conserved in pathogenic microorganisms. NLPs have two conserved domains in their amino acid sequence: two cysteines connected with a disulfide bond and a hepta-peptide (GHRHDWE), forming a negatively-charged cavity. These two elements are particularly essential for plant cell membrane damaging [32,40,41,42]. It has been further demonstrated that these NLPs can cause cell death and ethylene production during the interaction between dicotyledonous and fungi [40,41,42]. Nazar Pour et al. (2020) identified six homologs to NLP proteins in *N. parvum* that induced necrosis in tomato (*Solanum lycopersicum*), and the synthesis of these NLPs seemed to result from plant recognition by fungi, however, the mechanisms inducing necrosis remain to be elucidated [40]. Cobos and colleagues (2019) identified four NLP-like proteins in *D. seriata.* They demonstrated that these NLPs could play a role in the pathogenicity of *D. seriata*, inducing necrosis on the leaf margin that progressed through the center of the leaves of infected grapevine [32]. Interestingly, in the same study, the authors were able to note an improvement in the conductance of leaves where NLPs had been infiltrated. This phenomenon could be directly related to the loss in cell membrane integrity caused by NLPs [32]. Other authors had already raised the hypothesis that the location of NLPs near cell and nuclear membranes could induce a loss in membrane integrity [42]. Localization of NLPs into the cytoplasm and even into the nucleoplasm could be an explanation for the particularly toxic effect of these proteins. They could then interfere with genic transcription, induce cell death or interact with chloroplasts [32]. Indeed, Nazar Pour and colleagues (2020) had also noticed that NLPs from *N. parvum* harmed the photosynthesis of tomato leaves infiltrated with this kind of effector [40].

### 2.2. Carbohydrate-Active Enzymes Are Key Actors of Grapevine Colonization by Botryosphaeriaceae

Carbohydrate-active enzymes (CAZymes) are responsible for the biosynthesis, modification or breakdown of complex carbohydrates and glycoconjugates. They play an important role in fungal pathogen–host interactions, particularly plant cell wall-degrading enzymes (PCWDE), involved in the pathogenicity of fungi by degrading plant cell-wall polymers and allowing the release of sugars as carbon source. These enzymes can be intra- or extracellular, the latter being the most involved in the degradation of primary and secondary plant cell walls [27,43,44,45,46,47,48]. Compared to other Dothideomycetes, Botryosphaeriaceae genomes are richer in CAZymes-related genes [46,49,50], such as *Colletotrichum* and *Fusarium* species which share the same lifestyle plasticity and broad host range [10,25,33]. This high content of CAZymes in Botryosphaeriaceae genomes is closely similar to that of opportunistic fungi, more than pathogenic fungi [10]. According to the literature, the number of predicted CAZyme encoding genes is comprised between 504 and 753 in *N. parvum* [25,27,33,46,50,51], 504 in *N. mediterraneum* [27], between 485 and 825 in *B. dothidea* [27,33,50,51], between 465 and 789 in *L. theobromae* [10,27,33,50,52,53], between 433 and 577 in *D. mutila* [27,33], between 432 and 662 in *D. seriata* [27,33,46,50] and 395 in *D. viticola* [27]. Interestingly, 294 genes encoding CAZYmes carrying a signal peptide were identified in *N. parvum*, indicating a potential role in extracellular activities [51]. Researchers showed that the lifestyle and host diversity of fungi are associated with the CAZyme content of genomes, and that their aggressiveness is a result of a part of the latter, combined with other pathogenicity factors [25,27,33,48,54,55,56]. For example, it has been shown that *N. parvum* is one of the most aggressive Botryosphaeriaceae species on grapevines [27,57,58], in accordance with its high CAZyme content [27,33,50]. In contrast, *B. dothidea* is the second species containing the highest number of these genes, but shows a low aggressiveness on grapevines [27,57,58].

CAZyme classification is composed of five enzyme classes (glycoside hydrolases, GHs; glycosyltransferases, GTs; polysaccharides lyases, PLs; carbohydrates esterases, CEs; auxiliary activities, AA) and an associated module (carbohydrate-binding modules, CBMs) [43,59,60]. Among grapevine-pathogenic Botryosphaeriaceae, CAZymes comprise at least 54 GH families, 4 PL families, 9 CE and 9 AA families, 18 CBM families and 27 GT families, including, respectively, at least 20, 4, 5, 9 and 3 families belonging to the PCWDE (0 for the GT families) [3,25,27,33,46,50,51]. It has been reported that more than 34% of the putative secreted proteins of *N. parvum* and *D. seriata* genomes are enzymes involved in cell wall degradation, and the same is true for other grapevine trunk pathogens. Thus, the rich and diverse repertoire of CAZymes in Botryosphaeriaceae appears to be associated with their ability to colonize woody plants. Although CAZyme repertoire is shared by several pathogens associated with GTDs, a specific expression pattern of theses enzymes was evidenced after infection with fungi responsible for Esca (*Fomitiporia mediterranea*, *Stereum hirsutum*, *Phaeoacremonium chlamydospora* and *P. minimum*), Eutypa dieback (*Eutypa lata*) and Botryosphaeria dieback (*B. dothidea*, *D. seriata* and *N. parvum*). Therefore, it seems that CAZyme expression signature is characteristic of each pathosystem causing distinct symptoms [46,61]. The five classes (GHs, PLs, CEs, AAs and CBMs) targeting mainly celluloses, hemicelluloses, lignin and pectin are the most expanded in Botryosphaeriaceae genomes. These superfamilies are more enriched in *Botryosphaeria* and *Neofusicoccum* species than in other Botryosphaeriaceae such as *Diplodia*, *Dothiorella* and *Lasiodiplodia* [25,27,33,50]. Interestingly, depending on the *Botryosphaeria* strain, the most expanded CAZyme families target mainly hemicelluloses and lignin according to Wang et al. (PG45 and CBS 115476 *B. dothidea* strains) [25], while those described by Yu et al. [33] mainly degrade cellulose and pectin, showing the variability that can exist in this species.

Literature reports a variable number of PCWDE-coding genes within a genus. This can be explained by (i) the inter- and intraspecific diversity, and (ii) the classification width of CAZymes in PCWDE which can vary from 46 [33] to 52 [51] members. Considering the smallest classification of PCWDE-coding genes (i.e., 46 families), the genera *Neofusicoccum* and *Botryosphaeria* include the highest number of enzymes of interest, between 230 and 457, and between 235 and 356, respectively, followed by *Lasiodiplodia* (222–319), *Diplodia* (190–259) and *Dothiorella* (169–274). Among this classification, 9 families are the most expanded in these genera: AA9, GH31 and GH43, targeting cellulose and hemicelluloses, AA1, AA3 and AA7, targeting lignin, GH28, PL1 and PL3, targeting pectin, and CE5, targeting cutin [10,25,27,33,46,50,51]. Compared with other species within their genera, *D. seriata*, *L. theobromae* and *N. parvum* appeared as the species with the most PCWDE (with 243, 273 and 352 genes on average, respectively). On the other hand, *B. dothidea* (296), *D. mutila* (222), *D. viticola* (169) and *N. mediterraneum* (239) species are part of those with the less, representing for all these species between 42% and 56% of their CAZymes-coding genes [10,25,27,33,46,51].

Transcriptomic analysis of *L. theobromae* revealed that a lot of genes related to plant cell wall degradation are preferentially expressed in planta, especially in grapevine [10,62,63] and in cacao leaves [52], showing their role in fungi pathogenicity and plant colonization. To deepen the knowledge on *L. theobromae* virulence, Chethana et al. (2016) characterized the GH28 family by overexpression and silencing methods, and concluded that it induced an increase or a decrease of lesion lengths, respectively, after *L. theobromae* inoculation on grapevine shoots [64]. In the same way, a cutinase from *Botryosphaeria dothidea* (Bdo_10846) was shown to be involved in the fungus aggressiveness and in the development of wart symptoms on apple trees [65]. Finally, a study of *N. parvum* pathogenicity factors led to the identification of co-expressed gene clusters during grapevine infection that were enriched in both genes encoding PCWDEs and genes associated with secondary metabolism, suggesting a co-regulation that supports its aggressiveness. In addition, pairwise statistical testing of differential expression, followed by co-expression network analysis, revealed that physically clustered genes coding for putative virulence functions were induced depending on the substrate or stage of plant infection. Co-expressed gene clusters were significantly enriched not only in genes associated with secondary metabolism, but also in those associated with cell wall degradation, suggesting that dynamic co-regulation of transcriptional networks contributes to multiple aspects of *N. parvum* virulence. In total, 89 genes belonging to BCGs, as well as seven P450- and 20 PCWDE-encoding genes, were found to be expressed exclusively in planta [51].

### 2.3. Transporters Are Virulence Factors Enabling Fungal Nutrition and Resistance

Like many plant pathogenic fungi, Botryosphaeriaceae rely on transmembrane transporters to (i) assimilate sugars from plant wall degradation by CAZymes, as well as other nutrients such as amino acids or lipids, (ii) secrete secondary metabolites or compounds involved in pathogenicity or (iii) avoid accumulation of toxic compounds from the plant or anti-fungal treatments [66,67,68]. Most genomes in the Botryosphaeriaceae family are rich in genes belonging to the transporter family, a number of which are shared with species characterized as opportunistic [10]. In general, the number of cellular transporters varies from one species to another, according to the studies, from 2419 to 2490 in *L. theobromae* [10,27], from 1526 to 2588 in *N. parvum* [27,46,51], 2549 in *N. mediterraneum,* 2505 in *B. dothidea* and from 1345 to 2238 in *D. seriata* [27,46]. Similarly to *Colletotrichum* or *Fusarium* species, membrane transporters in the Botryosphaeriaceae family are found to be more abundant (over 917 in *L. theobromae*, being the richest in such transporters among Botryosphaeriaceae) than in other pathogenic fungi such as *M. oryzae*, *B. graminis* or *A. brassicicola* [10,52]. Furthermore, the transporter family is under positive or negative selection pressure regarding the species. The constant need to adapt to a changing environment is notably allowed by the evolution of transporter families. Thus, the gene families encoding transporters in species such as *L. theobromae* and *B. dothidea* show significant expansions, while *D. seriata* and *N. parvum* are contracting these families [27]. Interestingly, the low host specificity and broad geographical distribution of fungi from the Botryosphaeriaceae family such as *L. theobromae*, able to colonize various phylum such as Pinophyta, Magnoliids, Monocots and Dicots, indicate their ability to use different forms of nutrients [3,27,69,70].

Transporters can be categorized in nine classes including channels or pores that facilitate passive transport, primary active transporters ATP-driven and secondary active or electrochemical potential-driven transporters [67,71,72,73,74]. The latter category shows the highest number of genes across Botryosphaeriaceae species, representing up to 31 to 42% annotated transporters depending on the studies [27,46]. Among these, the major facilitator superfamily (MFS; transporter classification database (TCBD), 2.A.1) transport simple sugars, oligosaccharides, inositols, drugs, amino acids, nucleosides, organophosphate esters, Krebs cycle metabolites and a large variety of organic and inorganic anions and cations in solution in response to chemiosmotic ion gradients or proton motive force [75]. This family is found to be particularly enriched in the Botryosphaeriaceae family compared to other transporter families, as described for Ascomycetes in general [27,67], and represents between 455 and 514 predicted genes in the genera *Lasiodiplodia, Neofusicoccum* and *Botryosphaeria* where they are found to be more numerous than in other species from the Botryosphaeriaceae family [27,33,51,52]. In planta, studies evidenced up-regulation of *L. theobromae* and *N. parvum* genes from the MFS family in particular those transporting sugars during infection of grapevine, and sometimes as soon as 24 h post inoculation [51,62,63], as observed in other genera such as *Colletotrichum* [24,76,77]. Primary active transporters represent the second most abundant transporter category with up to 27% annotated transporters. Among these, the ATP-binding cassette (ABC) superfamily (TCDB, 3.A.1) gathers multicomponent transporters containing highly conserved amino acid motifs, capable of transporting across membranes, in response to ATP hydrolysis, both small molecules and macromolecules including toxins, metal ions, fatty acids and secondary metabolites [67,78,79]. The Botryosphaeriaceae genome known to have the highest number of genes encoding ABC-transporters is *Dothiorella sarmentorum* (134 genes compared to around 60 for other Botryosphaeriaceae species, similarly to *Colletotrichum* and *Fusarium*) [27,51,80]. During the infection of grapevine, four ABC transporters were up-regulated in *L. theobromae,* suggesting their role in fungal pathogenesis [62]. Within this class of transporters, the pleiotropic drug resistance (PDR) subfamily, that is described as providing a way to expel the drugs and associated with azole resistance, was up-regulated by *L. theobromae* during grapevine infection [10]. In addition to drug, toxin or phytotoxin resistance characteristics conferred by both MFS and ABC transporters to fungi, descriptions of their involvement in secondary metabolism are also provided in literature. These families are notably identified in several secondary metabolism gene clusters of *B. dothidea* and might be involved in the exportation of secondary metabolites [25].

### 2.4. Expansion of Genes Associated with Secondary Metabolism and Production of Highly Diversified Phytotoxic Metabolites

#### 2.4.1. Genomic and Transcriptomic Studies of Secondary Metabolic Pathways

Overall, analysis of all available Botryosphaeriaceae genomes show richness in secondary metabolite biosynthetic gene clusters compared to other Dothideomycete genomes [10,81]. Morales Cruz et al. (2018) identified 252 gene clusters involved in the synthesis and secretion of secondary metabolites in grapevine trunk pathogens, including Botryosphaeriaceae. The total number of genes related to secondary metabolism ranged from 142 in *D. seriata* to 353 in *N. parvum* [61]. Secondary metabolite synthesis probably represents an important virulence factor during wood colonization, since gene families specifically expanded in Ascomycete trunk pathogens were significantly enriched in genes associated with secondary metabolism [46]. The most abundant gene cluster in Botryosphaeriaceae was represented by polyketide synthases (PKS), followed by non-ribosomal peptide synthetases (NRPS), terpene synthases (TS) and PKS-NRPS hybrid clusters [51,81]. Polyketide synthases constitute an enzyme family responsible for the synthesis of polyketides from acyl CoA. Polyketides are a major class of naturally occurring secondary metabolites with diverse chemical structures and broad-spectrum functions. Non-ribosomal peptide synthetases generate non-ribosomal peptides from amino acids and are involved in the biosynthesis of peptide secondary metabolites. Finally, terpene synthases synthesize terpenes from activated isoprene units [82,83]. Recently, the number of PKS genes was evaluated in several Botryosphaeriaceae genera and revealed that the largest number (from 23 to 31) was reached in *Botryosphaeria*, *Lasiodiplodia*, *Neofusicoccum* and *Macrophomina* while *Diplodia* and *Dothiorella* contained the least numbers (11 and 8, respectively) [33].

Among gene families associated with secondary metabolism, [46] also found different classes of cytochrome P450 genes abundant in Ascomycete trunk pathogens such as *N. parvum* and *D. seriata*. They reported 212 and 122 genes encoding P450 in the genomes of *N. parvum* and *D. seriata,* respectively. Similarly, Yu et al. (2022) reported that Botryosphaeriaceae contained the largest number of genes encoding cytochrome P450, especially *N. parvum* with 267 genes [33]. In fungi, P450s are involved in housekeeping functions such as synthesis of essential membrane lipids and are also key players in the synthesis of secondary metabolites and detoxification of xenobiotic compounds [84]. In addition, expansion of P450 gene family could explain the wide host range of Botryosphaeriaceae fungi [33].

A recent transcriptome profiling of *N. parvum* virulence factor repertoire in the presence of grapevine wood showed that co-expressed gene clusters were enriched in genes associated with secondary metabolism [51]. Transcriptional dynamics during woody stem colonization further revealed that the activation of secondary metabolism takes place during later stages of infection since expression of genes associated with this pathway peaked only two weeks after inoculation. Co-expressed gene clusters belonged to the mevalonate pathway involved in the biosynthesis of sterols and terpenoids. They also comprised epoxide hydrolases involved in detoxication of plant aromatic compounds and in the synthesis of toxins [51].

#### 2.4.2. Diversity of Secondary Metabolite Synthesis

In accordance with the expansion of genes involved in secondary metabolite synthesis, Botryosphaeriaceae produce a high diversity of secondary metabolites considered as toxins. These toxins are thought to play an essential role in the fungi pathogenicity and virulence. Metabolites produced by *Neofusicoccum* species belong to different classes of natural products: 5,6-dihydro-2-pyrones, cyclohexenones, fatty acids, melleins, myrtucommulones, naphthalenones, naphtoquinones, phenols and alcohols and sesquiterpenes [85]. One could hypothesize that specific secondary metabolite profile could reflect different aggressiveness of various Botryosphaeriaceae strains. However, characterization of secreted secondary metabolites reported that different isolates of *N. parvum* have a similar profile. Structure elucidation showed that metabolites belonged to four different chemical families: dihydrotoluquinones (terremutin and derivatives), epoxylactones (asperlin and dia-asperlin) dihydroisocoumarins (mellein and hydroxymelleins) and hydroxybenzoic acids (methyl-salicylic acid hydroxypropyl salicylic acid) [29]. Mellein and hydroxymelleins were also isolated from culture filtrate from *D. seriata* and mellein was further detected in grapevine wood infected by *D. seriata* [86]. A number of secondary metabolites with high structure diversity are also produced by *L. theobromae*, belonging to diketopiperazines, jasmonates, lactones and mellein. In detail, *L. theobromae* synthesizes compounds belonging to cyclohexenes and cyclohexenones, depsidones, diketopiperazines, indoles, jasmonates, lactones, melleins, phenyl and phenol derivatives, 2-(2-Phenylethyl)chromones, phytohomones and preussomerins [69].

Among the different classes of metabolites, diverse free fatty acids and their esters were identified in cultures of Botryosphaeriaceae, including *L. theobromae*, *N. parvum* and *N. vitifusiforme* [87,88]. Fatty acids are precursors for many secondary metabolites involved in fungal virulence, such as jasmonates, and they also constitute a source of acetyl CoA for polyketide-type metabolites [69,87,89].

## 3. Various Effects of Botryosphaeriaceae Pathogenicity Factors on Grapevine Metabolism and Defenses

A number of studies have reported the induction of defense responses in different grapevine tissues (especially leaves and wood) after natural or artificial inoculation with Botryosphaeriaceae pathogens [47]. However, despite the activation of plant defense mechanisms, it seems that GTD associated fungi overcome these responses and efficiently colonize the wood. The subsequent paragraphs discuss about the action of Botryosphaeriaceae pathogenicity factors on grapevine, which could explain how the fungi finally prevail on persistent plant immunity, at the level of gene regulation, metabolite production, cell functions and symptom development.

### 3.1. Phytotoxic Effect of Secondary Metabolites and Small Proteins

In grapevine, Botryosphaeria dieback is associated with several symptoms such as wood vascular discoloration, vascular cankers, leaf chlorosis and bud necrosis. Since most of secondary metabolites produced by Botryosphaeriaceae exert a phytotoxic activity, these symptoms are likely the result of fungal toxin production. Indeed, since GTD fungi are never detected in leaves of diseased grapevines, it has been hypothesized that toxin production in wood and migration to leaves could be involved in the appearance of GTD leaf symptoms. In addition, different Botryosphaeriaceae species produce similar metabolite arsenal which might explain the symptom similarity induced by these fungi.

Mellein and its derivatives have been detected in most strains isolated from grapevines affected by trunk diseases. Phytotoxic activity of several toxins has been tested via necrosis induction on grapevine leaves or leaf discs. Mellein and its derivatives induce necrosis on grapevine leaf discs, with (R)-(−)-3-hydroxymellein being the most toxic compound [29]. Among cyclohexenones, the most active metabolites are cyclobotryoxide, terremutin and epi-sphaeropsidone, while among naphtalenones, botryosphaerone represents the principal phytotoxic compound [85,90].

Phytotoxic effect of toxins has been mainly studied in vitro and fungal toxins have been rarely identified in infected grapevines. However, terremutin and mellein were detected in the wood of grapevines with Botryosphaeria dieback symptoms, and therefore, these compounds are likely involved in wood discoloration and necrosis symptoms [29]. In addition, in a study investigating life traits of different Botryosphaeriaceae, [91] showed that *N. parvum* isolate PER20 virulence could be related to a specific metabolite profile, including terremutin and salicylic acid derivatives.

On the other hand, involvement of fungal toxins in GTD symptom expression was recently questioned. Reveglia et al. (2021) showed that mellein is detected in wood artificially or naturally inoculated with *D. seriata*, and that the toxin amount is correlated to the amount of pathogen quantified by qPCR. However, no translocation of mellein to grapevine tissues free of pathogen (leaves) was evidenced, showing that this metabolite may not be responsible for foliar symptom expression [92]. In another recent study, Trotel Aziz et al. (2022) characterized several *N. parvum* mutant strains, impaired in the production of mellein or terremutin. These mutants remained pathogenic on grapevine, showing that these two metabolites alone are not essential for fungal virulence. However, they could play a quantitative role in the infection process [93]. Apart from secondary metabolites, SSPs may also be involved in the induction of wood necrosis and foliar symptoms of GTD, as discussed in the above paragraph.

Overall, it is likely that the virulence of fungi associated to GTD results from a synergistic effect of all pathogenicity factors.

### 3.2. Manipulation of Plant Defense Pathways by Pathogenicity Factors

#### 3.2.1. Inhibition of Programmed Cell Death

In a recent study, Zhang et al. [30] identified 119 candidate effectors from *Botryosphaeria dothidea*. The effect of these effectors on plant cell death was investigated after transient expression with *A. tumefaciens* in the *N. benthamiana* non-host system. Most of the effectors (116) inhibited the programmed cell death induced by BAX1. Seven effectors were further studied and found to completely inhibit PCD triggered by the INF1, MKK1 and NPK1 elicitors. These results suggest that effectors from *B. dothidea* could suppress PAMP triggered immunity in order to escape from plant defense and efficiently colonize the host [30]. A disturbance on cytoskeleton early-stage re-arrangement and later strong induction on oxidation reduction process have been reported on *D. seriata* and *N. parvum* infected grapevine suspension cells [94]. Which were then confirmed by direct evidence on PCD marker genes in grapevine on tissue based single cell sequencing (Guan et al., manuscript preparing).

#### 3.2.2. Interaction with Plant Hormonal Pathways

Transcriptome analysis of *L. theobromae* in the presence of grapevine wood revealed an upregulation of genes involved in salicylic acid and phenylpropanoid degradation (encoding salicylate hydroxylase, intradiol ring cleavage dioxygenase and fumarylacetoacetase). This upregulation was enhanced by high temperature. Activation of L tyrosine catabolism pathway could lead to inhibition of SA pathway, favoring fungal development especially during heat stress [63].

Other than the inhibition of plant defensive pathways, Botryosphaeriaceae are known to produce fatty acids and jasmonates. Free fatty acids and jasmonates were identified in the culture filtrate from *L. theobromae*, *Neofusicoccum parvum* and *N. vitifusiforme* [69,87]. Linoleic acid and jasmonates could take part in signaling pathway during grapevine colonization by Botryosphaeriaceae. Indeed, a growth regulator activity on tobacco seedlings has been shown for fatty acid esters from *L. theobromae* [88]. On another hand, high jasmonate concentration is known to have an antagonistic effect on the SA pathway [95]. Hence it could be hypothesized that in addition to a toxic effect, jasmonates inhibit the SA pathway involved in stimulation of plant defenses. Downregulation of VvWRKY70, a putative central component of the SA pathway after treatment of *V. rupestris* cells with *D. seriata* secreted compounds [94] may be in accordance with this hypothesis. However, the respective contributions of SA and JA pathways in grapevine resistance/sensitivity to Botryosphaeriaceae are not fully elucidated since no plant mutant impaired in one or the other pathway is known. Nonetheless, resistance of apple to *B. dothidea*, the causing agent of apple ring rot would involve the SA pathway [96].

#### 3.2.3. Phytoalexin Degradation

Botryosphaeriaceae fungi involved in GTD are known to secrete enzymes involved in wood decay that could also participate in phytoalexin degradation. Indeed it has been reported that *N. parvum* and *D. seriata* are able to quickly metabolize resveratrol and δ-viniferin, two major stilbene phytoalexins of grapevine [31,97]. Interestingly, the highest metabolizing activity was measured for *D. seriata* in accordance with a high laccase activity measured in extracellular protein extracts in the presence of grapevine wood. These results suggest that phenolic compound decay activity could help GTD fungi to bypass defense responses of the host [97]. Other studies are in accordance with this hypothesis. Transcriptome analysis of *L. theobromae* during grapevine infection showed an upregulation of genes involved in plant phenylpropanoid precursor degradation encoding salicylate hydroxylase, tyrosinase, (homo)gentisate dioxygenase and fumarylacetoacetate hydrolases [62,63]. Massonnet et al. (2016) also reported an early upregulation of salicylate hydroxylase expression during woody stem colonization by *N. parvum*, suggesting that this fungus is able to overcome host immunity via SA and phenolic metabolization [51].

### 3.3. Reactions to Pathogenicity Factors at Cellular Level

The above discussed pathogenicity factors are aimed at the plant xylem, where formed by the dead cells such as vessels belonging to vascular tissue or living cells such as paratracheal parenchyma, fibers and rays (For review of the classic CODIT in grapevine, compartmentalization of decay in trees, see [98]). Therefore, the resource allocation could be subjected to a trade-off between vessel occlusion and defense reaction in conjunctive living tissues in grapevine stem. As well as the signaling transduction cascade within single cells and amongst the symplasts through plasmodesmata. Grapevine secondary cell walls can be degraded/impeded by substantial amounts of enzymes produced by GTD associated Botryosphaeriaceae. Pectin rich pit membranes connecting xylem vessels, as well as gels secreted in vessels can be degraded by compounds with pectinolytic activity, benefiting from already existing openings in cell walls. Pathogens can also spread out from a parenchyma cell [99]. PCWDEs can act both intra- or extracellularly, the latter being the most involved in the degradation of primary and secondary plant cell walls [27,43,44,45,46,47,48]. In contrast, grapevine cell wall thickening in paratracheal parenchyma impede lateral hyphae penetration in xylem parenchyma cells, and cell wall modifications in living fibers and ray parenchyma is associated with suberin rather than lignin deposits. Suberized layers in ray parenchyma could efficiently restrict the fungus spread from one fascicular portion to another [100]. Cells infected by *L. theobromae* are described with reduced hardness, less materials for cell wall establishment, covalent/ion soluble pectin, fibers, but with higher value of water soluble pectin and cell wall polysaccharide-disassembling enzymes [101]. Other pathogenicity factors such as cutinases were also described as involved in virulence of *B. dothidea*, the plant infected by this pathogen showed a prompt increasement of phellem layer where the infecting hyphae sieged, together with a newly formed secondary cork layer and callose formation [65].

Recently, based on the classic zigzag model, studies using suspension cell lines and callus cells to consider the cells in a naïve system can simplify the model focusing on the host responses from detection of transcription and metabolites. After the outermost defense barrier was broken through by the pathogens, it is the plasma membrane that act as the primary site in the symplast, where pathogen-associated molecular patterns (PAMPs) are perceived. Ca^2+^ signals are diverse, depending on plant species and process specificity in PTI and ETI. A transient (within minutes) Ca^2+^ elevation has been reported on grapevine and other species ([102]; Guan et al., manuscript preparing). Subsequently, microtubule rearrangement, calcium signal amplification, phospholipid signaling and ethylene signaling have been reported to be triggered by Botryosphaeriaceae pathogen [94,103,104]. During wounding, some hormone signals, e.g., ethylene, have been showed to regulate occlusion of vessels via tylose development, constituting a barrier for fungal spread in the plant [105]. The interdisciplinary research utilizing the biotechnology such as single-cell sequencing and technics from material science such as quasi-in situ characterization can meet the above goal to develop new insights in certain tissues in stems affected by casual agents of GTD.

## 4. Conclusions

This review evidenced the recent advances in the deciphering of pathogenicity factors of Botryosphaeriaceae associated with GTD. The highlighted researches indicate that the classes of genes encoding small secreted proteins, enzymes associated with carbohydrate degradation, genes associated with secondary metabolism and transporters (closely related to the two last classes) are involved in the pathogenicity of Botryosphaeriaceae. Furthermore, the differential combination of these different factors contributes to explain the differential aggressiveness of species within this family affecting grapevines. Next generation sequencing provided tremendous amounts of data that help to better understand the pathosystem at a big scale, but further functional analyses will enable to better understand the specific role of candidate pathogenicity factors.

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
