# Peer review of "Pathogenicity Factors of Botryosphaeriaceae Associated with Grapevine Trunk Diseases: New Developments on Their Action on Grapevine Defense Responses"

_pathogens, 2022, doi:10.3390/pathogens11080951_

Round 1

Reviewer 1 Report

Grapevine, on one side plays important role as one of the most economic crops globally, on the other side suffers from various diseases. Trunk disease whose symptoms mainly in the branches comparing to those diseases whose symptoms mainly in leaves e.g. the widely studied powdery mildew are with different pathogen infection and plant defense mechanisms. In this study, the authors focus on the pathogenicity factors of Botryospaeriacea, summarized the recent findings on small secreted proteins, carbohydrate associated enzymes, transporters and secondary metabolism within the Botryosphaeriaceae family and their roles during grapevine infection. Meanwhile the grapevine defense mechanisms are reviewed, especially from the view of the cell and tissue illustration, which is interesting for further study. This review integrates the recent knowledges from plant and fungi, which will illuminate the ideas for scientists from both sides to get an overview of the plant-pathogen interaction zig-zag model. There are some points can be improved.

1.      In the abstract “This review highlights the knowledge gained on genes encoding small secreted proteins such as effectors, carbohydrate associated enzymes, transporters and genes associated to secondary metabolism, their distribution and evolution within the Botryosphaeriaceae family”. In fact, the information on “distribution and evolution" topic is not much in the text.

2.      The title “on plant metabolism” is not accurate, the defense response is far more than metabolism, may be replaced by the key word “during plant defense”

3.      Plant small signaling peptides (SSPs) are a class of plant polypeptides with special functions including pathogen response. In line77, what is the relationship between SSP (small secreted protein) and SSP (small signaling peptides)?

Author Response

Dear reviewer,

we would like to thank you for your careful reading of our manuscript and for the sound advices proposed. The corrections are directly included in the manuscript (highlighted in yellow). Concerning the remark 3, SSPs as described in the text correspond to secreted proteins present in fungi of all phylogenetic groups. They are characteristically shorter than 300 amino acids in length and have a signal peptide. The description can be found in this article https://www.frontiersin.org/articles/10.3389/fmicb.2020.00455/full.

Best regards.

Reviewer 2 Report

Dear Authors

This review highlights the knowledge gained on genes encoding small secreted proteins such as effectors, carbohydrate-associated enzymes, transporters, and genes associated with secondary metabolism, their distribution and evolution within the Botryosphaeriaceae family, and their expression during grapevine infection. Finally, it highlights the need for further research to elucidate the pathogenicity mechanisms of the Botryosphaeriaceae family of fungi infecting grapevine in order to improve the development of control methods and varietal resistance and to reduce the development and the effects of the disease on grapevine harvest quality and yield.

Line 16: importance and causing

Line 20: carbohydrate-associated

Line 20: associated with secondary

Lines 43-44: Add the following sentence “Grapevine, whose physiology and composition of the berries is the subject of various studies [17-19], has been the most economically impacted plant of agronomic interest in recent decades….”

[17] Cataldo, E., Salvi, L., Paoli, F., Fucile, M., & Mattii, G. B. (2021). Effects of defoliation at fruit Set on vine physiology and berry composition in cabernet sauvignon grapevines. Plants, 10(6), 1183.

[18] Cataldo, E. C., Salvi, L. S., Paoli, F. P., Fucile, M. F., Masciandaro, G. M., Manzi, D. M., ... & Mattii, G. B. M. (2021). Effects of natural clinoptilolite on physiology, water stress, sugar, and anthocyanin content in Sanforte (Vitis vinifera L.) young vineyard. The Journal of Agricultural Science, 159(7-8), 488-499.

[19] Tyagi, K., Maoz, I., Lapidot, O., Kochanek, B., Butnaro, Y., Shlisel, M., ... & Lichter, A. (2022). Effects of gibberellin and cytokinin on phenolic and volatile composition of Sangiovese grapes. Scientia Horticulturae, 295, 110860.

Line 226: Colletotrichum and Fusarium

Line 278: 2.4.1. Genomic and transcriptomic studies of secondary metabolic pathways

Line 319: 2.4.2. Diversity of secondary metabolite synthesis

Line 395: 3.2.1. Inhibition of programmed cell death

Line 408: 3.2.2. Interaction with plant hormonal pathways

Line 430: 3.2.3. Phytoalexin degradation

Author Response

Dear reviewer,

we would like to thank you for your careful reading of our manuscript. We directly included in the manuscript the proposed corrections (highlighted in green). Concerning the items noted line 278, 319, 395 and 408, we do not understand exactly the points that need to be corrected, we used the automatic numbering in Word for these titles.

Best regards.